# The effect of colour on reading performance in children, measured by a sensor hub: From the perspective of gender

Tamara Jakovljević [1] *, Milica M. Janković[2], Andrej M. Savić[2], Ivan Soldatović[3], Ivan Mačužić[4], Tadeja Jere Jakulin[5], Gregor Papa[6], Vanja Ković[7]

**1** Sensor Technologies, Jožef Stefan International Postgraduate School, Ljubljana, Slovenia, **2** School of Electrical Engineering, University of Belgrade, Belgrade, Serbia, **3** Institute of Medical Statistics and Informatics, Faculty of Medicine, University of Belgrade, Belgrade, Serbia, **4** Faculty of Engineering, University of Kragujevac, Kragujevac, Serbia, **5** FTŠ Turistica, UP, Portorož, Slovenia, **6** Jožef Stefan Institute, Ljubljana, Slovenia, **7** Laboratory for Neurocognition and Applied Cognition, Faculty of Philosophy, University of Belgrade, Belgrade, Serbia

* tamara.jakovljevic@hotmail.com

## Abstract

In recent decades reported findings regarding gender differences in reading achievement, cognitive abilities and maturation process in boys and girls are conflicting. As reading is one of the most important processes in the maturation of an individual, the aim of the study was to better understand gender differences between primary school students. The study evaluates differences in Heart Rate Variability (HRV), Electroencephalography (EEG), Electrodermal Activities (EDA) and eye movement of participants during the reading task. Taking into account that colour may affect reading skills, in that it affects the emotional and physiological state of the body, the research attempts to provide a better understanding of gender differences in reading through examining the effect of colour, as applied to reading content. The physiological responses of 50 children (25 boys and 25 girls) to 12 different background and overlay colours of reading content were measured and summarised during the reading process. Our findings show that boys have shorter reading duration scores and a longer Saccade Count, Saccade Duration Total, and Saccade Duration Average when reading on a coloured background, especially purple, which could be caused by their motivation and by the type of reading task. Also, the boys had higher values for the Delta band and the Whole Range of EEG measurements in comparison to the girls when reading on coloured backgrounds, which could reflect the faster maturation of the girls. Regarding EDA measurements we did not find systematic differences between groups either on white or on coloured/overlay background. We found the most significant differences arose in the HRV parameters, namely (SDNN (ms), STD HR (beats/min), RMSSD (ms), NN50 (beats), pNN50 (%), CVRR) when children read the text on coloured/overlay backgrounds, where the girls showed systematically higher values on HRV measurements in comparison to the boys, mostly with yellow, red, and orange overlay colours.

**Data Availability Statement:** All relevant data are within the manuscript and S1 Data.

**Funding:** The Authors work were supported through the Slovenian Research Agency (research

core funding No. P2-0098), AD Futura Fund (Public Scholarship, Development, Disability and Maintenance Fund of the Republic of Slovenia, and the Ministry of Education, Science and Technological Development of the Republic of Serbia. The funders had no role in study design, data collection and analysis, decision to publish, or preparation of the manuscript.

**Competing interests:** The authors have declared that no competing interests exist.

## Introduction

In children reading skills progress over developmental stages [1], while learning to read is one of the most important achievements of the early school years [2]. The complex process of reading skill acquisition involves perception and cognition, via integration of auditory and visual information processing, and memory, attention, and language skills [3]. Reading skills depend upon a range of cognitive abilities and perceptual processes that affect learning during development [4–6]. Gender differences in cognitive abilities during development have been widely analysed in neuropsychological and psychological research. However, these differences are still subject to debate [7–11]. Some studies have reported such differences [12, 13], while others report that they are not able to isolate them [14–18]. This implies the need for additional research on gender differences during cognitive development. More generally, in the past decade, the question of whether males and females differ in cognitive ability has been the focus of significant research [19]. While males and females do not differ in general intelligence, which is a general consensus [20], gender differences are commonly observed for more specific cognitive abilities such as visual-spatial ability [21] and language [22]. However, most gender differences are small or trivial (close to zero) in magnitude, explained by the Hyde gender similarities hypothesis (GSH) [15]. The gender gap in reading achievement, which is found cross-culturally, may be one exception to this hypothesis [19, 23, 24]. Hyde [25] concluded that it is "difficult to reconcile" the magnitude of the gender gap observed in reading with that in other domains of verbal ability, which is typically much smaller, as is claimed by Linn [26].

In the developmental context, girls tend to be superior to boys in verbal abilities and linguistic function from infancy through to adulthood, and for this reason gender difference has been investigated in previous studies, reporting greater reading achievements in girls [24, 27], with boys being better in visuospatial tasks involving memory [28]. Female students read more frequently and had a more positive attitude towards reading, resulting in better reading comprehension [29]. Recent research reports a larger Scan Path Length and Saccade Amplitude in female subjects [30]. Also, in tests of early reading ability Harper and Pelletier [31] found no gender differences in children's performance. However, the study employing eye-tracing methodology revealed gender differences in reading abilities indicated by Saccade Duration, Regression Rate, and Blink Rate [27].

Currently there are very few sources that explore the impact of colour on the reading process, specifically with early school-age children [32, 33]. The influence of text-, background-, or overlay-colour on the reading process in children is reported in literature [34–36] but there is no clear consensus regarding this. A recent study reported that colour does not influence the reading process [37], while conversely another has found that colour may be particularly effective for early readers such as school-age children [38]. In the study of visual stress, it is found that male subjects prefer blue and green, and females prefer pink and purple overlay colours for reading [39]. Therefore, the effect of colour on the reading process from the perspective of gender is interesting for further investigation.

Reading involves attention, memory, and sensory integration, which may be reflected in the psycho-physiological state of the individual engaged in the reading task. These processes are a result of fundamental physiological and neural processes, which are measurable by different BioSignal modalities such as Electrocardiography (ECG), Electroencephalography (EEG), Electrodermal Activity (EDA), and eye movement. The goal of the recent study was to incorporate multimodal sensor measurements to investigate the effect of colour on the content within the reading task in children from the perspective of gender differences. We have included measurements of heart rate variability (HRV), EEG, EDA, and eye-tracking to assess the influence of background and overlay colour on reading performance in boys and girls

attending the early years of primary school. We aimed to address the mechanisms of colour effect on the reading process through electrophysiological correlates of the reader's state while taking into account the gender aspect of the reading acquisition.

The current research is an attempt to present new evidence regarding gender differences in reading skill. This research aims to contribute to the existing body of knowledge on the effect of the text, background and overlay colour according to gender.

We aimed to investigate the effects of colour on the content as a stressor during the reading task. The present study aims to further illuminate underlying physiological and behavioural processes accompanying the reading task in children from the gender perspective.

## Materials and methods

### Participants

The study was carried out with fifty healthy participants, boys and girls (25 plus 25) randomly chosen from students in the second and third years (aged 8–10) of primary school "Drinka Pavlović" in Belgrade. Inclusion criteria were that children have normal or corrected-to-normal vision, and no reading and learning disabilities or attention disorders. A typical case for exclusion would be the presence of large artefacts in the acquired signals. However, no such cases were observed in our sample. According to these criteria, no participants were excluded from the statistical analysis.

Children individually participated in the experiment, each under the same experimental conditions: during the school day and in the same small classroom. They received a short instruction about the experiment setup. After they finished the reading test, participants received a certificate and a small present from the researcher. The research process was anonymous and the collected data were anonymised. Only the data regarding the gender and age of the participants were available to the research team. Before starting the experiment, researchers received oral informed consent about the students' participation from the parents at a school class meeting, organised by the school director and the class teacher, which was summarized in the school director's note and delivered to researchers.

The ethical committee of the Psychology Department of the University of Niš (a branch of the Serbian Psychology Association) approved the experimental procedure No 9/2019.

### Procedure

A computer screen and keyboard were placed in front of every child. The participants read the text in silence from the computer screen, as per the instructions from the researcher at the beginning of the experiment. Each of the participants read the story from the stimuli presentation on the screen, pressing the space button to receive the next paragraph on the next slide. The experiment started with the presentation of black text on white background (the referent slide) as children would typically see in daily life. After that, a pseudo-randomised background colour with black text was presented to the children along with an overlay version (marked by O in the further text, e.g. "red O is for red overlay") of the presented slides. The text on each colour/slide was different but was kept in a logical order. Except for the referent slide, no other colour was fixed to any particular stimuli presentation. There are no other factors (semantic or affective content, syntax, vocabulary, or text complexity) that could impact the reading process apart from the actual colour, given that the colours were randomly presented to the participants, rather than being selected.

The experiment design was exactly the same as in [40].

Fig 1 shows the sensor hub which consists of a portable multimodal ECG/EEG/EDA and eye tracking system for physiological data acquisition during the reading task. For data real-

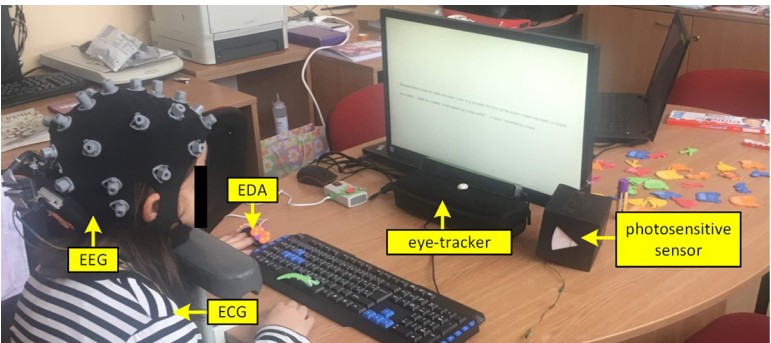

**Fig 1. Portable multimodal ECG/EEG/EDA system and eye-tracking system.**

time monitoring and storage two laptops were used, one for the ECG/EEG/EDA system, and one for eye movement signal monitoring (connected with an external keyboard and the screen positioned in front of the child).

A portable remote eye tracker (SMI RED-m 120-Hz, https://www.smivision.com) was mounted in front of the participants and fixed to the screen to secure its stability. In order to ensure that each participant was the same distance from the screen, an adjustable chin-rest was mounted on the table (it was placed 16 cm above the table and 57 cm from the eye-tracking sensor) [41]. For the stimuli presentation SMI Experiment Centre 3.7 was used. An iView RED-m was used for data storage and collection.

The Smarting (mBrainTrain, Belgrade, Serbia) mobile system with a 24-channel EEG amplifier was used for recording EEG and ECG signals, which were communicating wirelessly with a laptop via Bluetooth. In the experiment a Greentek Gelfree-S3 cap with twenty-two monopolar EEG channels was used (10/20 locations: Fp1, Fp2, F3, F4, C3, C4, P3, P4, O1, O2, F7, F8, T7, T8, P7, P8, Fz, Cz, Pz, AFz, CPz, POz). The FPz electrode was used as the ground site and for the reference site the FCz electrode was used. The ECG signal was recorded by one channel of the Smarting amplifier using a surface SKINTACT ECG electrode placed at the left chest region above the heart. The signals of EEG and ECG were acquired with 250 Hz sampling rate and 24-bit resolution. Prior to the test, the skin-electrode impendence was below the manufacturer's recommended value of 1 kOhm.

For the synchronisation of the multimodal ECG/EEG/EDA and the eye tracking system, one channel of the Smarting amplifier with a small photosensitive sensor was used. This sensor registered the colour changes between two subsequent slides (one black and one white slide lasting for 200 ms each, positioned between two presented slides) indicating the colour changes of the presented slides.

A custom-made galvanic skin response device [41] that sends data via Bluetooth to a laptop was used (sampling rate 40 Hz) for electrodermal activity (EDA) acquisition. EDA data were recorded on the laptop using the Smarting application.

## Data processing

BeGaze 3.7 software was used to monitor eye tracking data. Eye tracking analysis included the following parameters: a) Fixation Count, b) Fixation Frequency (count/second), c) Fixation Duration Total (ms), d) Fixation Duration Average (ms), e) Saccade Count, f) Saccade Frequency (count/second), g) Saccade Duration Total (ms) and i) Saccade Duration Average (ms).

EEG/ECG/EDA data were analysed using Matlab ver. 8.5 (Mathworks, USA) in the following manner for each presented slide:

1. EEG data of all subjects was processed offline. EEG signals were band-pass filtered using 4th order Butterworth filter to extract the activity in the following frequency bands: Delta (0.5–4 Hz), b) Theta (4–7 Hz), c) Alpha (7–13 Hz), d) Beta (15–40 Hz) and e) broadband EEG activity (0.5–40 Hz).

   Filtered signals of all subjects/channels were squared and segmented according to the event markers while each epoch was associated with the reading task of one slide. The median value of data associated with each epoch was calculated for obtaining a single band-power value. Median calculation is used to remove impulse-noises associated with movements, blinks and other artefacts that may occur in the EEG during reading within each epoch. Additional visual inspection of power epochs was conducted to ensure that the median values represent the valid quantification of the band-power activity of each epoch.

2. Heart activity beats were extracted using Kubios HRV Premium 3.3.1. software [42, 43]. The beats are detected using the Kubios built-in algorithm based on the Pan–Tompkins algorithm [44]. The period between two beats, so called beat-to-beat interval (BBI), and time domain heart rate variability (HRV) parameters [45], Table 1, were extracted by the same software. Also, the Kubios built-in threshold based artefact correction algorithm was performed (a local average interval of 0.35 s was selected and the detected artefacts were automatically replaced by cubic spline interpolated values within the software).

3. The average value of EDA data was calculated for each slide.

## Statistical methodology

Here we present results as percentages, means ± standard deviation or taking into account data type and distribution. We compared groups (boys vs. girls) using a parametric test, an independent samples t-test. All p-values which were less than 0.05 were considered significant. The data were analysed within the SPSS 20.0 software (IBM Corp. Released 2011. IBM SPSS Statistics for Windows, Version 20.0. Armonk, NY: IBM Corp.). The Bonferroni corrections were applied in all the statistical analysis where necessary as a control for multiple comparisons.

## Results

### Reading results on white (default) background with black text

Gender comparisons (girl vs. boys) regarding the examined parameters for white background only are presented in Table 2. A significant difference has been obtained regarding a single HRV parameter for pNN50 (%), where girls have higher scores in comparison to boys. In all other parameters of EEG frequency bands (Alpha, Beta, Theta, Delta), ECG parameters and Eye tracking measurements, we observed no significant difference between girls and boys.

**Table 1. HRV parameters.**

| Parameter (Unit) / Time domain parameters | Description |
|---|---|
| Mean RR (ms) | Mean value of BBIs |
| SDNN (ms) | Standard deviation of normal BBIs |
| Mean HR (beats/min) | Mean value of heart rate |
| STD HR (beats/min) | Standard deviation of heart rate |
| CVRR = SDNN/Mean RR (n.u.) | Coefficient of variance of normal BBIs |
| RMSSD (ms) | Root mean square of differences of successive BBIs |
| NN50 (beats) | Number of successive BBIs that varied more than 50 ms |
| pPNN50 (%) | Percentage of successive BBIs that differ more than 50 ms |

**Table 2. Reading duration, EEG, eye tracking, EDA and HRV parameters in girls and boys—significant p values are marked as bold.**

| Parameters | Grade | | p value* |
|---|---|---|---|
| | **MALE (n = 25)** | **FEMALE (n = 25)** | |
| **Reading duration** | | | |
| RD (s) | 40.32± 21.64 | 49.04± 23.49 | 0. 21 |
| **EEG parameters (median power band)** | | | |
| Alpha | 12.64± 8.47 | 11.99± 6.56 | 0.76 |
| Beta | 5.50± 3.00 | 5.77± 2.89 | 0.75 |
| Delta | 133.64± 198.80 | 81.32± 52.95 | 0.21 |
| Theta | 20.31± 28.69 | 16.19± 9.54 | 0.50 |
| Whole Range | 134.57± 76.02 | 130.77± 76.62 | 0.86 |
| **Eye tracking parameters** | | | |
| Fixation Count | 39.88± 21.48 | 37.68± 15.40 | 0.69 |
| Fixation Frequency [count/s] | 1.02± 0.45 | 0.97± 0.52 | 0.74 |
| Fixation Duration Total [s] | 48.29± 47.98 | 44.30±22.18 | 0.72 |
| Fixation Duration Average [ms] | 1,120.83± 588.25 | 1,201.85± 554.00 | 0.63 |
| Saccade Count | 34.68± 11.23 | 32.32± 14.59 | 0.53 |
| Saccade Frequency [count/s] | 0.95± 0.45 | 0.86± 0.52 | 0.53 |
| Saccade Duration Total [ms] | 784.50±289.72 | 722.84±361.63 | 0.52 |
| Saccade Duration Average [ms] | 22.49± 3.75 | 22.38± 5.59 | 0.94 |
| **EDA value** | | | |
| EDA (uS) | 7.66± 3.71 | 7.69± 3.61 | 0.98 |
| **HRV parameters** | | | |
| Mean RR (ms) | 664.09± 54.58 | 673.95± 99.37 | 0.67 |
| SDNN (ms) | 40.37± 19.12 | 54.34± 36.03 | 0.09 |
| CVRR (n.u.) | 0.07±0.03 | 0.08±0.04 | 0.20 |
| Mean HR (beats/min) | 90.93± 7.43 | 90.76± 12.36 | 0.95 |
| STD HR (beats/min) | 5.51± 2.35 | 6.89± 3.09 | 0.08 |
| RMSSD (ms) | 48.75± 29.40 | 69.97± 56.33 | 0.10 |
| NN50 (beats) | 11.60±11.75 | 20.32±17.63 | 0.05 |
| pNN50 (%) | 22.76±17.75 | 36.78±25.63 | **0.03** |

Independent sample t test

## Background and overlay colours

In Table 3 a comparison between girls and boys, based on the t-test for independent samples, was obtained on each of the parameters measured in the study, namely: Reading duration, EEG, Eye tracking, EDA and HRV. As is obvious from the table, the girls (coloured in red) scored systematically higher in many of the HRV measurements. In particular for SDNN (ms) they scored higher on yellow, red O, orange O, and purple O; for CVRR they scored higher on yellow, red O, yellow O, orange O and purple O; for STD HR girls scored higher on red, yellow, orange, turquoise, red O, blue O, yellow O, orange O and purple O; for RMSSD they scored higher on yellow, turquoise, red O, yellow O, orange O and purple O; for NN50 they scored higher on red, yellow, orange, turquoise, red O, blue O, yellow O, orange O and turquoise O; and for pNN50 they scored higher on yellow, orange, purple, turquoise, red O, yellow O, orange O and purple O. Boys only scored higher when reading on a purple background for the following eye-tracking parameters: Saccade Count, Saccade Duration Total and Saccade Duration Average (coloured in blue).

**Table 3. Differences between girls (marked with red colour) and boys (marked with blue colour) on reading duration, EEG, eye tracking, EDA and HRV parameters ($p < .05$).**

| Parameters | Normalized values | | | | | | | | | | | |
|---|---|---|---|---|---|---|---|---|---|---|---|---|
| | red | blue | yellow | orange | purple | turquoise | red O | blue O | yellow O | orange O | purple O | turquoise O |
| **Reading duration** | | | | | | | | | | | | |
| RD (s) | | | | | | | | | | | | |
| **EEG parameters (median power band)** | | | | | | | | | | | | |
| Alpha | | | | | | | | | | | | |
| Beta | | | | | | | | | | | | |
| Delta | | | | | | | | | | | | |
| Theta | | | | | | | | | | | | |
| Whole Range | | | | | | | | | | | | |
| **Eye tracking parameters** | | | | | | | | | | | | |
| Fixation Count | | | | | | | | | | | | |
| Fixation Frequency [count/s] | | | | | | | | | | | | |
| Fixation Duration Total [s] | | | | | | | | | | | | |
| Fixation Duration Average [ms] | | | | | | | | | | | | |
| Saccade Count | | | | | ▪(blue) | | | | | | | |
| Saccade Frequency [count/s] | | | | | | | | | | | | |
| Saccade Duration Total [ms] | | | | | ▪(blue) | | | | | | | |
| Saccade Duration Average [ms] | | | | | ▪(blue) | | | | | | | |
| **EDA value** | | | | | | | | | | | | |
| EDA (uS) | | | | | | | | | | | | |
| **HRV parameters** | | | | | | | | | | | | |
| Mean RR (ms) | | | | | | | | | | | | |
| SDNN (ms) | | | ▪(red) | | | | ▪(red) | | | ▪(red) | ▪(red) | |
| CVRR | | | ▪(red) | | | | ▪(red) | | ▪(red) | ▪(red) | ▪(red) | |
| Mean HR (beats/min) | | | | | | | | | | | | |
| STD HR (beats/min) | ▪(red) | | ▪(red) | ▪(red) | | ▪(red) | ▪(red) | ▪(red) | ▪(red) | ▪(red) | ▪(red) | |
| RMSSD (ms) | | | ▪(red) | | | ▪(red) | ▪(red) | ▪(red) | ▪(red) | ▪(red) | | |
| NN50 (beats) | ▪(red) | | ▪(red) | ▪(red) | | | ▪(red) | ▪(red) | ▪(red) | ▪(red) | | ▪(red) |
| pNN50 (%) | | | ▪(red) | ▪(red) | ▪(red) | ▪(red) | ▪(red) | ▪(red) | ▪(red) | ▪(red) | ▪(red) | |

Girls vs. boys across all of the examined parameters over averaged scores aggregated for all tested colours are presented in Table 4. Boys achieved higher scores on a few EEG and eye-tracking measurements, namely: Delta and Whole range EEG band measurements and Fixation Count, Saccade Count and Saccade Duration Total. The girls, on the other hand, scored higher on the Reading Duration and on a few HRV measures, namely: SDNN, STDHR, RMSSD, NN50, PNN50 and CVRR.

## Discussion

The Reading Duration, EEG, eye tracking, EDA and HRV parameters were evaluated in 50 children (25 female and 25 male second and third year students (aged 8–10) of primary school) using a multimodal sensor hub. As reading process involves attention, memory, and sensory integration, which may be reflected in the psychophysiological state of the individual engaged in the reading task, the study aim was investigating different BioSignal modalities such as ECG, EEG, EDA, and eye movement during the reading task.

Gender differences in reading are widely reported [2, 19, 24, 27, 46–49], and it was found that motivation, attitudes, and the type of reading task could impact on reading skills in boys

**Table 4. Reading duration, EEG, eye tracking and EDA parameters in girls and boys across all colours together—significant p values are marked in bold.**

| Parameters | Grade | | p value* |
|---|---|---|---|
| | **MALE (n = 25)** | **FEMALE (n = 25)** | |
| Reading duration | | | |
| RD (s) | 41.83± 22.61 | 48.83± 27.72 | **0.00** |
| EEG parameters (median power band) | | | |
| Alpha | 11.02± 6.06 | 10.50± 5.94 | 0.27 |
| Beta | 5.32± 2.61 | 5.50± 3.82 | 0.50 |
| Delta | 82.90± 81.16 | 61.04± 38.66 | **0.00** |
| Theta | 15.31± 14.16 | 13.98± 8.35 | 0.14 |
| Whole Range | 116.90± 69.03 | 103.82± 58.33 | **0.01** |
| Eye tracking parameters | | | |
| Fixation Count | 39.55± 22.15 | 35.79± 11.74 | **0.01** |
| Fixation Frequency [count/s] | 1.01± 0.48 | 1.00± 0.82 | 0.81 |
| Fixation Duration Total [s] | 47.49± 39.36 | 45.22±26.61 | 0.41 |
| Fixation Duration Average [ms] | 1,154.48± 534.75 | 1,196.31± 554.87 | 0.34 |
| Saccade Count | 34.95± 15.10 | 30.75± 10.01 | **0.00** |
| Saccade Frequency [count/s] | 0.93± 0.46 | 0.88± 0.87 | 0.33 |
| Saccade Duration Total [ms] | 776.14±441.89 | 667.28±246.73 | **0.00** |
| Saccade Duration Average [ms] | 21.76± 3.27 | 22.01± 5.27 | 0.48 |
| EDA value | | | |
| EDA (uS) | 7.56± 3.24 | 7.83± 3.64 | 0.31 |
| HRV parameters | | | |
| Mean RR (ms) | 656.20± 51.22 | 662.65± 90.05 | 0.26 |
| SDNN (ms) | 40.08± 16.87 | 52.59± 29.66 | **0.00** |
| Mean HR (beats/min) | 92.00± 7.31 | 92.10± 11.36 | 0.89 |
| STD HR (beats/min) | 5.46± 1.99 | 6.90± 2.47 | **0.00** |
| RMSSD (ms) | 44.89± 23.32 | 65.14± 46.05 | **0.00** |
| NN50 (beats) | 12.25±10.57 | 21.51±18.19 | **0.00** |
| pNN50 (%) | 21.45±16.03 | 34.23±24.02 | **0.00** |
| CVRR | 0.07±0.03 | 0.09±0.03 | **0.00** |

*independent sample t test

more closely than in girls [47]. It was speculated that the boys' reading performance could depend more on their motivation and attitude. The results of the present study showed that boys had shorter reading duration parameters than girls, but at the same time, they scored higher in some eye- tracking measures, and had longer Fixation Count, Saccade Count, and Saccade Duration Total measurements than the girls, irrespective of background/overlay colour. They also had a longer Saccade Count, Saccade Count Total and Saccade Count Average when reading text on a purple background. Additionally it is reported that males have a more positive emotional response than females during competitive game play [50]. Therefore, here we have taken into account findings that suggest that boys are more motivated to read in new conditions, without teachers' grades/assessments, and have more competitive attitudes in comparison to female students. It has also been reported that male students have poor reading abilities in comparison to girls [27, 51], and therefore make more exploratory eye movements, which consequently result in larger Saccade Amplitudes as we have also demonstrated in the present research.

Regarding the normal maturation processes reflected in the EEG, McCarthy reported [52] that gender differences are equally distributed, while other researchers [53, 54] found that EEG differences between boys and girls suggest earlier maturation in girls [55]. On the other hand, Cohn [56] and Gasser [57] found no differences between boys and girls measurable by EEG. It is also reported that the amount of activity in the lower frequency EEG bands decreases with age, and in higher frequency bands, it increases [58, 59]. The gender differences in EEG are also reported in context of the task performance and cognitive activity. During the task and rest phase, females have a higher EEG power than males [60, 61]. Also, gender differences are frequently reflected in numerous factors, like task and age [62]. Some research shows that in most cognitive tasks including language, there exist inappreciable differences in behavioural output between the genders [16–18]. EEG power and its distribution over a lifetime also varies between the genders [63]. EEG maturation markers increase in faster band activity (alpha, beta) and decrease in slower band activity (theta, delta) [55, 57, 64–66]. In the present study it is shown that boys have higher values of delta band and whole range of EEG in comparison to girls. Clarke et al. report that boys' EEG matures faster than girls' in childhood, but that these differences are eliminated during adolescence. Conversely Gasser et al. [57] declare that there are no gender differences, mostly because of high interindividual variability in the EEG power spectrum. Other authors have found that until the age of 16 the alpha rhythm does not mature [67]. In previous research the delta band was found to be higher in young individuals than in adults because of incomplete cortical maturation, and is typically even higher in children with learning disabilities [68]. These findings are in line with our results, which showed increased Delta and Whole Range EEG bands to be more prominent in boys than in girls, which could be as a result of the faster maturation process in girls. This is in keeping with previously mentioned results showing that boys had a less mature pattern of eye-movements in comparison to girls.

Likewise, electrodermal activity has been used in several studies with the objective of clarifying markers of psychophysiological functioning and children's developmental processes [69, 70].

Several studies supported higher levels of baseline SCL (Skin Conductance Level) [68] and SCL reactivity to stressors [71] in girls in comparison to boys. In the present study we found no evidence of systematic differences between boys and girls for this measurement when reading text on either white or coloured/overlay background.

Physiological mechanisms during adolescence actively and progressively undergo changes. It has been reported that HRV progressively reduces with age, and development during adolescence can be assessed using the heart rate variability (HRV) [72]. HRV can be used to ascertain the evolution of the ontogenetic maturation [73–76]. Moreover, the gender influence measured by HRV parameters was manifested only in young adults and younger adolescents and our study group belongs to the same age category. Research study shows that measurements of HRV depend on the age but not on the gender of healthy children [77]. When different colours of background, text, and overlay were included in the reading process we found significant differences between girls and boys, whereby girls scored higher on HRV parameters. This indicates higher emotional reactions in girls when they read the text on the coloured/overlay background in comparison to the boys. Regarding this result, our findings are compatible with previously reported results showing that girls have higher values on SDNN and RMSSD measurements.

## Conclusion

Primarily this research aimed to assess gender differences in the reading process and to contribute to existing research on the effect of text, background and overlay colour according to gender.

Secondly, the aim was also to investigate the effects of colour on the content as a stressor in the reading task. In order to shed light on contradictory reports regarding gender differences in reading skills, present study illuminates underlying physiological and behavioural processes in the reading task in children from the gender perspective. It evaluates differences in reading duration, EEG, ECG, EDA and eye movement measures on both white and 12 different background/text/overlay colours. It was found that boys show shorter reading duration parameters than girls, and at the same time longer eye-tracking measures such as Fixation Count, Saccade Count, and Saccade Duration Total while reading on a coloured background/overlay, whereas they had a Longer Saccade Duration, Saccade Duration Total, and Saccade Duration Average on a purple background. These results partially support our expectation that boys would have more difficulties reading the text when displayed on background/overlay colours. However, they did not have more issues reading on the coloured background in comparison to the white/default background.

Comparing EEG parameters in girls and boys during reading on white background we did not find systematic differences. Observing all the colours together, it is shown that boys have higher values in Delta and Whole Range bands in comparison to the girls. As the Delta Range is higher in young adults, the findings are aligned with previous research where it is shown that boys will have more difficulties in reading tasks because the reading process is still not automated in comparison to the girls. In fact, they have also demonstrated longer Saccade Count and Saccade Duration measurements in comparison to the girls when reading on a purple background. It seems that the colour can really increase the task difficulty for less proficient readers. We did not find systematic differences for EDA measures between boys and girls while reading on white or coloured background/overlay content. However, regarding ECG measures, girls scored significantly higher on HRV measures (SDNN (ms), STD HR (beats/min), RMSSD (ms), NN50 (beats), pNN50 (%), CVRR), in particular on yellow, orange O and red O colours. These findings are also contributable to studies where it is shown that girls have higher values on HRV measures than boys, which is particularly evident from results including the additional effect of colour on the reading process.

Finally, we can underline that colours used as a stressor in a particular reading task could illuminate gender differences, especially in eye-tracking and ECG measures. Boys have shown longer Saccade Count and Saccade Duration in comparison to girls while reading on the purple colour. Boys have shown shorter reading duration than girls on all coloured background/overlay, and longer eye-tracking measures such as Fixation Count, Saccade Count, and Saccade Duration Total. Regarding the ECG (SDNN, STD HR, RMSSD, NN50, pNN50, CVRR) measures, girls scored higher than boys while reading on yellow, orange O, and red O colours. These findings show that colours could be contributing to a better understanding of gender differences and their relation to the context of the reading processes.

## Supporting information

**S1 Data.**
(XLSX)

## Acknowledgments

The authors acknowledge Elementary School "Drinka Pavlović" (Belgrade) and IPS Jozef Stefan.

## Author Contributions

**Conceptualization:** Tamara Jakovljević, Milica M. Janković, Tadeja Jere Jakulin, Gregor Papa, Vanja Ković.

**Formal analysis:** Milica M. Janković, Andrej M. Savić, Ivan Soldatović, Vanja Ković.

**Investigation:** Tamara Jakovljević.

**Methodology:** Tamara Jakovljević, Tadeja Jere Jakulin, Vanja Ković.

**Project administration:** Tamara Jakovljević.

**Resources:** Ivan Mačužić.

**Supervision:** Ivan Soldatović, Tadeja Jere Jakulin, Gregor Papa, Vanja Ković.

**Writing – original draft:** Tamara Jakovljević.

**Writing – review & editing:** Milica M. Janković, Andrej M. Savić, Gregor Papa, Vanja Ković.

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
