## [Decision Letter · Decision Letter 0]

23 Feb 2021

PONE-D-20-40129

Effect of colours on reading performance in children measured by the sensor hub: from the perspective of gender

PLOS ONE

Dear Dr. Jakovljevic,

Thank you for submitting your manuscript to PLOS ONE. After careful consideration, we feel that it has merit but does not fully meet PLOS ONE’s publication criteria as it currently stands. Therefore, we invite you to submit a revised version of the manuscript that addresses the points raised during the review process.

Please carefully address all the reviewers comments in the revised version in a point by point basis and elaborate your justification in detail. 

We look forward to receiving your revised manuscript.

Kind regards,

Murugappan M, Ph.D

Academic Editor

PLOS ONE

Journal Requirements:

2. You indicated that you had ethical approval for your study.

In your Methods section, please ensure you have also stated whether you obtained informed consent from parents or guardians of the minors included in the study or whether the research ethics committee or IRB specifically waived the need for their consent.

If you obtained parental consent, please also state whether your ethics committee or IRB approved the consent procedure.

3. Thank you for including your ethics statement: 

"The ethical committee of the Psychology Department of the University of Niš approved the experimental procedure.

No. 9/2019".   

a. Please provide additional details regarding participant consent.

In the ethics statement in the Methods and online submission information, please ensure that you have specified (i) whether consent was informed and (ii) what type you obtained (for instance, written or verbal, and if verbal, how it was documented and witnessed). If your study included minors, state whether you obtained consent from parents or guardians. If the need for consent was waived by the ethics committee, please include this information.

'The authors acknowledge the financial support from the Slovenian Research Agency (research core funding No. P2-0098), AD Futura Found (Public Scholarship, Development, Disability and Maintenance Found of the Republic of Slovenia), IPS Jozef Stefan and the Ministry of Education, Science and Technological Development of the Republic of Serbia.'

'No

The funders had no role in study design, data collection and analysis, decision to publish, or preparation of the manuscript.'

5. We note that Figure 1 includes an image of a participant in the study. 

As per the PLOS ONE policy (http://journals.plos.org/plosone/s/submission-guidelines#loc-human-subjects-research) on papers that include identifying, or potentially identifying, information, the individual(s) or parent(s)/guardian(s) must be informed of the terms of the PLOS open-access (CC-BY) license and provide specific permission for publication of these details under the terms of this license.

Please download the Consent Form for Publication in a PLOS Journal (http://journals.plos.org/plosone/s/file?id=8ce6/plos-consent-form-english.pdf). The signed consent form should not be submitted with the manuscript, but should be securely filed in the individual's case notes.

Please amend the methods section and ethics statement of the manuscript to explicitly state that the patient/participant has provided consent for publication: “The individual in this manuscript has given written informed consent (as outlined in PLOS consent form) to publish these case details”.

Reviewers' comments:

Reviewer's Responses to Questions

**Comments to the Author**

1. Is the manuscript technically sound, and do the data support the conclusions?

Reviewer #1: Partly

Reviewer #2: Partly

Reviewer #3: No

2. Has the statistical analysis been performed appropriately and rigorously? 

Reviewer #1: Yes

Reviewer #2: Yes

Reviewer #3: No

3. Have the authors made all data underlying the findings in their manuscript fully available?

Reviewer #1: Yes

Reviewer #2: Yes

Reviewer #3: No

4. Is the manuscript presented in an intelligible fashion and written in standard English?

Reviewer #1: Yes

Reviewer #2: Yes

Reviewer #3: No

5. Review Comments to the Author

Reviewer #1: Comments to the author:

This study aims to investigate the gender differences in reading achievement, cognitive abilities and maturation process using HRV, EEG, and EDA. The study is interesting, and the paper is well-written. There are however some major areas that require the authors' attention.

My major concern is about processing of bio signals. Generally, the physiological signals are affected by noise (e.g., power line interference) and artifacts (body movements). Especially, when you record signals from children. I was unbale to find the details about signal pre-processing. How the noise and artifacts are removed? How the authors extracted the band power? What filter used to extract the EEG sub-frequency bands? Overall, no clear information about pre-processing.

Next, I was unable to see the information about data analysis i.e., data segmentation or the authors analyzed the whole recorded signals. state the outcome of this research in real time? What is the significant of this research? Include the limitations

Reviewer #2: The research article analyzes various physiological and eye gaze parameters to understand the gender differences in reading comprehension of elementary students. The differences is observed in many of the parameters and concurs with existing research. Colors do not seem to influence children with good proficiency.

Some comments

1. What is the novelty proposed in this research work? How are the differences in controversial reports filled in this work?

2. Studies on the influence of color on reading can be understood by different questionnaire based methods and statistical analysis of the same. Was any such study done or feedback obtained from the teachers to validate your results. Is there a need to use invasive methods such as EEG?

3. Why do you think that the delta waves show significant differences?

4. What would the proposed application of these finding. Proposing a few may provide more insight into the paper.

Reviewer #3: The problems statement is good. Some of the results are presented but the signal processing aspects are missing.

Major comments

1. The paper is poorly written. Please check typos and rewrite the paper in standard English

2. Signal processing techniques are missing in this paper. It should be described clearly with mathematical expressions, algorithms with optimal coding parameters and also results.

6. PLOS authors have the option to publish the peer review history of their article (what does this mean?). If published, this will include your full peer review and any attached files.

Reviewer #1: No

Reviewer #2: No

Reviewer #3: **Yes: **M Sabarimalai Manikandan

---

## [Author Response · Author response to Decision Letter 0]

22 Apr 2021

Dear Reviewers, 

Thank you for your detailed and useful comments. We hope that you will find our answers satisfactory and we definitely feel that your comments helped us improve the manuscript significantly. 

Sincerely, 

Authors of the manuscript

Reviewer #1: Comments to the author:

This study aims to investigate the gender differences in reading achievement, cognitive abilities and maturation process using HRV, EEG, and EDA. The study is interesting, and the paper is well-written. There are however some major areas that require the authors' attention.

My major concern is about processing of bio signals. Generally, the physiological signals are affected by noise (e.g., power line interference) and artifacts (body movements). Especially, when you record signals from children. I was unabale to find the details about signal pre-processing. How the noise and artifacts are removed? How the authors extracted the band power? What filter used to extract the EEG sub-frequency bands? Overall, no clear information about pre-processing.

Next, I was unable to see the information about data analysis i.e., data segmentation or the authors analyzed the whole recorded signals. state the outcome of this research in real time? 

Response: 

We have included more detailed information about signal processing conducted within this study, such as type of filter, calculation of band-power, signal segmentation and inspection for artifacts and their removal. Fourth order Butterworth filter was used to extract the activity of 5 frequency ranges analyzed in this study. Median values of band-power were calculated for each epoch (slide)/ frequency band in order to obtain a single power value for the total reading duration of each slide. The choice of median power calculation was introduced in order to remove impulse-noises associated with movements, blinks and other artefacts which was stated in the text. Additional visual inspection of band-power values for each subject/band/epoch was conducted in order to validate the obtained median values and check for presence of artifacts.

Our study is of exploratory nature and includes offline data processing only for examining the gender differences during reading task (with different colour/overlay setups) using multimodal signal measurements. Future studies and analyses of the collected data may include exploration of optimal signal measurement setup which could in real time estimate the preferred colour/overlay in order to facilitate the reading task, but this is out of the scope of the current study. You can find more in Methods section (Page 5).

What is the significant of this research? Include the limitation

Response:

Regarding the significance of this research -  the second reviewer in his first question also asked about it and about the novelty of the results, so hopefully you will find our answer satisfying. 

One of the limitations was that we could have developed a longitudinal study where we would follow a group of girls and boys for a number of years, but this was outside of the scope of the current study and exceeds our current resources. 

Reviewer #2: The research article analyzes various physiological and eye gaze parameters to understand the gender differences in reading comprehension of elementary students. The differences is observed in many of the parameters and concurs with existing research. Colors do not seem to influence children with good proficiency.

Some comments

1. What is the novelty proposed in this research work? How are the differences in controversial reports filled in this work?

Response:

The novelty and significance of this research is primarily methodological - we attempted to develop a sensory hub in order to simultaneously test cognitive and emotional arousal in boys and girls whilst reading on different colour/overlay backgrounds in order to overcome a pure behavioural measurements and get a more fine-grained insight into the processes and differences involved in process of reading across the two groups. The differences observed in controversial reports may have to do with strategic responding to some extent (given that the behavioral responding does not capture the underlying differences and strategies in the task), which we hopefully overcame by applying an automated way of measuring both cognitive and emotional responses. 

2. Studies on the influence of color on reading can be understood by different questionnaire based methods and statistical analysis of the same. Was any such study done or feedback obtained from the teachers to validate your results. Is there a need to use invasive methods such as EEG?

Response:

Yes, we actually informally interviewed teachers as well as some speech specialists in the field regarding reading on the coloured background/overlay backgrounds, and they suggested that this kind of intervention may be of a great help in focusing attention in children whose attention span is much shorter nowadays. We did not run a separate study based on the feedback, but it become clear to us that, apart from looking into group differences, like in this study, the future research will need to focus more on the individual differences, as it seems that kids tend to have their own prefered colour and that choice vary significantly. But, that would be a matter of some future research, as it exceeds the scope of the article and results we presented here. 

We ensure you, as we did teachers and parents of the children who took part in this study, that none of the methodologies we used here is invasive to children in any way. We literally told them that it is measuring brain waves and not changing them, in the same way as we can measure body temperature without changing it. The amount of infra-red light used in the eye-tracking system is definitely such that it can not cause any harm to childrens’ eyes. Otherwise, we, as researchers, would have a huge ethical dilemma, even before getting a formal ethical approval for this study (which we did). Also, kids felt that they took part in a scientific adventure and they loved it - to the extent that they would come back to the researchers during the break between classes.  

3. Why do you think that the delta waves show significant differences?

Response:

This is a really interesting question. In our opinion the delta waves show significant differences because boys get to mature later in comparison to girls, and previous research (55, 70) demonstrated that younger (less matured children and children with learning disabilities)  tend to have more prominent delta waves. In our research we also found that boys had a less mature pattern of eye-movements as we interpreted/described in the discussion.  

4. What would the proposed application of these finding. Proposing a few may provide more insight into the paper.

Response:

One would be that we can make the reading process easier by selecting and adjusting the colour of the background for each individual child. The second would be that by applying this system we could help prevention, but also early detection of potential problems that children may have with reading. The major plan would be to extend this stream of research to dyslexic children and to employ machine learning in order to be able to do clear adjustment for each individual child. 

Reviewer #3: The problems statement is good. Some of the results are presented but the signal processing aspects are missing.

Major comments

1. The paper is poorly written. Please check typos and rewrite the paper in standard English

Response:

Thank you so much - you were absolutely right. We asked a professional and also a native speaker of English to do the necessary corrections and there were so many of them that we did not keep track-changes of them, because the text would be difficult to read and major corrections difficult to spot. We also intend to get our language expert to make a final read before the paper gets published, if you find all the changes we made satisfactory. 

2. Signal processing techniques are missing in this paper. It should be described clearly with mathematical expressions, algorithms with optimal coding parameters and also results.

Response:

We have used the Kubios HRV Premium 3.3.1software for the heart beat extraction and beat-to-beat interval calculation, as well as for time domain HRV parameters calculation. We have properly referenced this software (added two references: [42], [43]) and we have modified the corresponding paragraph in the text (Page 5):

“2) heart activity beats were extracted using Kubios HRV Premium 3.3.1. software [42, 43]. The beats are detected using the Kubios built-in algorithm based on the Pan–Tompkins algorithm [44]. The period between two beats, so called beat-to-beat interval (BBI), and time domain heart rate variability (HRV) parameters [45], Table 1, were extracted by the same software. Also, the Kubios built-in threshold based artefact correction algorithm was performed (for local average interval was selected 0.35 s and the detected artefacts were automatically replaced by cubic spline interpolated values within the software).”

[42] https://www.kubios.com/hrv-premium/ 

[43] Tarvainen, M. P., Niskanen, J. P., Lipponen, J. A., Ranta-Aho, P. O., & Karjalainen, P. A. (2014). Kubios HRV–heart rate variability analysis software. Computer methods and programs in biomedicine, 113(1), 210-220.

---

## [Decision Letter · Decision Letter 1]

19 May 2021

The effect of colour on reading performance in children, measured by a sensor hub: from the perspective of gender

PONE-D-20-40129R1

Dear Dr. Jakovljevic,

We’re pleased to inform you that your manuscript has been judged scientifically suitable for publication and will be formally accepted for publication once it meets all outstanding technical requirements.

Kind regards,

Murugappan M, Ph.D

Academic Editor

PLOS ONE

Additional Editor Comments (optional):

I may strongly suggest the authors to address the reviewer 3 suggestions in the camera-ready version of the manuscript.

Reviewers' comments:

Reviewer's Responses to Questions

**Comments to the Author**

1. If the authors have adequately addressed your comments raised in a previous round of review and you feel that this manuscript is now acceptable for publication, you may indicate that here to bypass the “Comments to the Author” section, enter your conflict of interest statement in the “Confidential to Editor” section, and submit your "Accept" recommendation.

Reviewer #1: All comments have been addressed

Reviewer #2: All comments have been addressed

Reviewer #3: All comments have been addressed

2. Is the manuscript technically sound, and do the data support the conclusions?

Reviewer #1: Yes

Reviewer #2: Yes

Reviewer #3: Partly

3. Has the statistical analysis been performed appropriately and rigorously? 

Reviewer #1: Yes

Reviewer #2: Yes

Reviewer #3: No

4. Have the authors made all data underlying the findings in their manuscript fully available?

Reviewer #1: Yes

Reviewer #2: Yes

Reviewer #3: Yes

5. Is the manuscript presented in an intelligible fashion and written in standard English?

Reviewer #1: Yes

Reviewer #2: Yes

Reviewer #3: No

6. Review Comments to the Author

Reviewer #1: The authors addressed all the reviewer comments. I think the paper can be accepted for publication.

Reviewer #2: The authors have addresses the comments. The data collection methods have been justified. The authors have also included signal processing steps for better understanding.

Reviewer #3: Please improve the writing and presentation of this paper.

I would suggest the authors to highlight their statistical findings in the conclusion.

7. PLOS authors have the option to publish the peer review history of their article (what does this mean?). If published, this will include your full peer review and any attached files.

Reviewer #1: No

Reviewer #2: **Yes: **Jerritta Selvaraj

Reviewer #3: No

---

## [Editor Report · Acceptance letter]

26 May 2021

PONE-D-20-40129R1 

The effect of colour on reading performance in children, measured by a sensor hub: from the perspective of gender 

Dear Dr. Jakovljevic:

I'm pleased to inform you that your manuscript has been deemed suitable for publication in PLOS ONE. Congratulations! Your manuscript is now with our production department. 

Kind regards, 

on behalf of

Dr. Murugappan M 

Academic Editor

PLOS ONE